# Comparing virtual reality, desktop-based 3D, and 2D versions of a category learning experiment

**Robin Colin Alexander Barrett**[1], **Rollin Poe**[2], **Justin William O'Camb**[1], **Cal Woodruff**[3], **Scott Marcus Harrison**[2], **Katerina Dolguikh**[2], **Christine Chuong**[4], **Amanda Dawn Klassen**[2], **Ruilin Zhang**[2], **Rohan Ben Joseph**[3], **Mark Randall Blair**[1,2]*

1 Department of Psychology, Simon Fraser University, Burnaby, British Columbia, Canada, 2 Cognitive Science Program, Simon Fraser University, Burnaby, British Columbia, Canada, 3 Department of Computing Science, Simon Fraser University, Burnaby, British Columbia, Canada, 4 Department of Statistics and Actuarial Science, Simon Fraser University, Burnaby, British Columbia, Canada

* mblair@sfu.ca

**Data Availability Statement:** Data is available on the Federated Research Data Repository (FRDR) at the following https://doi.org/10.20383/102.0539.

## Abstract

Virtual reality (VR) has seen increasing application in cognitive psychology in recent years. There is some debate about the impact of VR on both learning outcomes and on patterns of information access behaviors. In this study we compare performance on a category learning task between three groups: one presented with three-dimensional (3D) stimuli while immersed in the HTC Vive VR system (n = 26), another presented with the same 3D stimuli while using a flat-screen desktop computer (n = 26), and a third presented with a two-dimensional projection of the stimuli on a desktop computer while their eye movements were tracked (n = 8). In the VR and 3D conditions, features of the object to be categorized had to be revealed by rotating the object. In the eye tracking control condition (2D), all object features were visible, and participants' gaze was tracked as they examined each feature. Over 240 trials we measured accuracy, reaction times, attentional optimization, time spent on feedback, fixation durations, and fixation counts for each participant as they learned to correctly categorize the stimuli. In the VR condition, participants had increased fixation counts compared to the 3D and 2D conditions. Reaction times for the 2D condition were significantly faster and fixation durations were lower compared to the VR and 3D conditions. We found no significant differences in learning accuracy between the VR, 3D, and 2D conditions. We discuss implications for both researchers interested in using VR to study cognition, and VR developers hoping to use non-VR research to guide their designs and applications.

## Introduction

The emergence of accessible, affordable virtual reality (VR) technology in the last decade provides both an opportunity and an imperative for cognitive scientists who study learning and attention. The opportunity is methodological, as VR enables the controlled presentation of a

**Funding:** The author(s) received no specific funding for this work.

**Competing interests:** The authors have declared that no competing interests exist.

variety of novel experimental situations that would not normally be possible in a laboratory environment, while also making it possible to collect multiple varieties of behavioral data. Immersive virtual reality technologies have already seen significant interest with diverse applications, including investigations of cognitive phenomena such as the color contrast illusion [1], episodic memory [2, 3], and selective attention [4]. This technology has also made it possible to conduct neurological studies while exposing participants to a wider variety of stimuli than possible with regular flat computer screens used in most brain-imaging experiments [5–7]. The imperative for researchers manifests from the growing excitement over the use of this emerging technology. While designers have been eager to implement tasks in VR, our understanding of how VR does or does not influence cognitive processing is underdeveloped, and direct comparisons of findings from existing experimental paradigms are needed to discover whether findings and theories from extant work apply to the unique affordances of virtual reality experiences. The present work looks specifically at learning and information access, two cognitive processes that the existing literature, as discussed below, suggests might be impacted by a VR implementation. We note that there have been a variety of technologies that have been referred to as "virtual reality" in the scientific literature including CAVE Systems [8–10], Fish Tank systems [11], and Dextroscopes [12, 13]; in the present discussion of VR, we will be limiting our discussion to the use of immersive virtual reality devices which use head-mounted displays and motion-tracked hand controllers to immerse the user into virtual environments, such as the Oculus Rift, HTC Vive, and Valve Index.

While much of the research already produced using VR has been conducted in relation to industry and education relevant measures such as learning outcomes [8, 14, 15], very little attention has been given to the actual cognition that takes place while using these devices. Regardless, there is much in the literature that provides both direct and indirect evidence of what we might expect in this medium, and so the next section of this paper approaches the issue from two main angles. Firstly, we examine the evidence that although the tasks implemented are highly similar to their real-world counterparts the underlying cognitive processes and subjective experiences are distinct. Next, we point to evidence to the contrary where, despite being a new experience and interface, the cognitive processes which manage the interaction are unchanged. After reviewing the literature related to VR, we turn to the category learning paradigm specifically, the central focus of the research reported. We briefly introduce the field, summarizing typical findings in this area, showcasing recent developments using eye tracking to detect learning-related changes in information access behaviors. Using the literature of this field, we present a case for what we might expect to see in a VR implementation of this task. No prior work known to the authors has ever examined category learning in the context of a VR interface. Comparing information access behaviors and learning performance in a typical category learning experiment to behavior observed in a VR implementation of this type of experiment is then the focus of the empirical work described in this paper.

## Same tasks, different experiences

There is a fundamental distinction in the sensory and perceptual properties of VR environments and those of other presentation methods. Instead of being "in the distance", objects perceived in virtual environments are physically presented on a screen which sits within just a few centimeters from the eyes [16]. Subsequently, while accommodation (focal distance within the eye) and convergence (distance from the eyes at which both eyes are foveated) are coupled and coincident in the real world, VR displays require that these be decoupled and divergent. Current VR headsets also have a restricted field of view, requiring more frequent repositioning of the head and body to access information, rather than simply being able to make eye

movements to desired areas of interest. This difference alone in display characteristics requires that users to spend more time acquiring information required for any particular task as they must make more physical movements to perceive the same amount of information that would be available freely to someone working in the real world with no constraints to their field of view. There is also some lag between the user's self-motion and the resulting optical transformation of the display, screen shape and/or display optics that can lead to subtle image distortions; and there may be conflict between virtual visual cues and real-world vestibular cues [16, 17]. All of these differences are inherent to the nature of the medium itself, and therefore, even if the environments being experienced are identical, the physical qualities of the medium create a unique experience for VR users that is distinct from the experience of reality.

VR also differs from usual lab conditions. The physical mode of presentation in VR replaces the entire visual field and moves to match the head movements of its user with any accompanying audio presented through headphones, blocking out potential distractors from the real-world space and making for a very different experience than the computer monitor setups typically used to present stimuli in experiments. More immersive setups have been shown to result in a greater feeling of presence [18, 19], defined in the literature as a participant's subjective sense of "being there" within a simulation [18]. Feeling a higher degree of presence while immersed in VR has been reported to result in higher feelings of engagement with learning tasks than traditional media [10, 15, 20–22], but whether or not this will have an impact on actual measures of interest such as learning outcomes is largely contested [23–26]. Regardless, it is still worth noting that in the context of academic research, having more engaged participants in an experiment might perhaps lead to less participant fatigue, providing researchers with better quality data.

Makransky and colleagues [19, 27, 28] consistently find that immersive VR groups often experience a higher cognitive load while performing VR simulation tasks in comparison to other methods of instruction, likely extending somewhat from the unfamiliarity of the VR interface, as well as the design of the virtual environment. They find that this increase in cognitive load often leads to a decline in learning outcomes. However, this same research group also finds that VR-based instruction, when paired with other evidence-based pedagogical techniques such as generative learning [23, 29], enactment [30], or pre-training [27], do much better than non-immersive learning groups in terms of declarative and procedural learning outcomes.

While it is possible to use a VR headset for immersive visual input coupled with standard interaction methods such as a keyboard and mouse or a standard gamepad, part of the appeal of VR is surely the use of tracked hand controllers that allow for more natural grasping and object manipulation that matches the realism of the immersive 3D environment. This interaction method has extra costs in time and motor requirements however, as maneuvering through a complex 3D environment and moving one's arms to interact with that environment is undoubtedly more effortful than navigating and interacting with a desktop computer or pen-and-paper learning apparatus. There are competing ideas as to how these motor costs might interact with cognition and learning-related behaviors, however. On the one hand, the use of VR may constitute a desirable difficulty [31, 32], wherein the added cost required to access information leads to increased learning outcomes and faster optimization of information access behaviors over time. Work exploring information access costs in attention research discussed in the next section expands this idea further. On the other hand, this motor cost may also be framed as an increase to cognitive load [33, 34] during a learning task. In Zhao et al. [35], participants navigated a virtual reality environment in VR while another group performed the same task on a desktop computer. When testing their memory of landmarks encountered within the virtual environment, their results found no advantage of VR over the

desktop-based simulation. Instead, when asked to correctly remember the direction of different landmarks, some advantages were found favoring the desktop condition. While this study was not framed by its authors as a difference in information access costs, the fact remains that the motor costs inherent to VR-based navigation may have increased cognitive load in participants, potentially contributing to the findings observed.

Overall, researchers have provided good reasons to anticipate that results of studies utilizing VR-based materials might produce different results than would be typical in more traditional mediums of research, and conversely, that findings and patterns of behavior from traditional lab research of human cognition may not yield accurate templates for behavior of VR users.

## New interface, same cognitive processes

Although the previous section suggests that the differences in implementation, both in terms of immersion and information access cost, might impact learning outcomes and information access behaviors, evidence can also be mustered to support the opposing view. A review of VR studies in education by Hamilton et al. [36] examined quantitative measures of learning outcomes, including test scores, completion times, and retention measures. Out of the twenty-nine studies reviewed, ten showed no differences in learning outcomes between VR-based learning and other Desktop-based learning programs, while two showed worse learning outcomes for the VR conditions in at least some comparisons. The authors concluded that while in many cases there were measurable improvements in learning outcomes when using VR-based learning programs, this improvement was not guaranteed. Similarly, a recent meta-analysis by Angel-Urdinola et al. [14] reported an overall positive effect of VR on learning outcomes compared to desktop-based learning and pen-and-paper instruction, but also report that more than half of the studies cited were neutral-positive; VR had no impact on learning beyond what was possible with traditional learning methods. In one case, Parong & Mayer [21] found that 2D presentation slide-based learning initially outperformed a 12-minute VR experience for learning biology concepts. In a follow-up experiment, students were given the chance to reflect at regular intervals throughout the VR experience to better make up for the fact that participants in the slide-based condition were able to stop and pause whenever they needed. Scores in the VR condition improved, eventually to be at an equal level to the participants who learned from paper slides [21]. While none of these studies address underlying cognitive processes involved in learning, they do suggest that the quality of one's learning is not tied only to the modality it is learned in, with the result that test scores are not likely to dramatically go up or down just because content is delivered in VR instead of through a textbook or desktop computer.

VR-based assessments of cognitive performance have been found to have convergent and discriminant validity with real-world versions of the same tasks, including the Stroop task [37], the Posner task [4, 38], the Humphrey Perimeter Examination [39], and various tests of learning and memory [40]. In Pugnetti et al. [41], a VR-based test of memory was compared to pen-and-paper analogues, and it was found that for the VR group, the likelihood of recall was greater for episodic items (such as a pedestrian involved in a near-miss) than operational, which were in turn more likely to be remembered than task-irrelevant items. The researchers speculated that VR scenarios, by emulating scenarios that better resemble the real world, could provide more nuanced data about real world memory performance than traditional pen-and-paper tests. They reason that VR systems could engage both episodic and semantic memory systems, enhancing recall of certain items. Likewise, VR exposure therapy for acrophobia has been shown to produce clinically relevant improvements on measures of anxiety and avoidance [42], showcasing that the experiences had in VR are similar enough for anxiety to be

reduced in the real world for people who first receive exposure to the object of their phobia in VR. Taken together, these findings reinforce the imperative for researchers to employ VR in a wider variety of research areas to see where this technology can continue to bring increased validity to lab-based experiments.

Despite the physical differences in user interfaces between VR and other computer methods frequently used during learning, users might still follow the same patterns of information access behaviors across instantiations. Comparing data from e-sports replay files to data collected from prior eye tracking experiments, McColeman et al. [43] showed that expertise-related changes in patterns of information access behaviors were similar, regardless of whether information access was mediated by eye-movements on a static screen or by hand movements on a mouse and keyboard. Furthermore, patterns of information access behaviors were found to be similar between entirely different tasks: a simple category learning experiment structure and the video game StarCraft 2 [44]. Working from a more neurological perspective, Reggente et al. [45] reviewed research combining fMRI with immersive head mounted VR displays, and reported that areas of the brain traditionally associated with spatial navigation in the real world were also active during VR-based tasks in many of the studies reviewed. These findings suggest that information access and learning interact in predictable ways regardless of changes in the physical task interface. Eichert et al. [46] considered whether language-driven anticipatory movements recorded in regular eye tracking research could be seen in the behavior of participants using VR, and again found that findings from previous research replicated well in the VR environment. Overall, these findings have prompted a great deal of optimism that models of cognition formed in laboratory settings will increase in ecological validity as improvements in VR allow for the presentation of increasingly realistic scenarios within the controlled space of a lab, without fundamentally altering the nature of the cognitive processes involved due to the use of immersive VR.

## Category learning

The category learning paradigm has experimental antecedents predating the cognitive sciences as a whole [47, 48]. Having been an active area of research for over a century, category learning provides a set of well documented methodological implementations and stable findings [43, 49, 50]. It asks participants to classify visual stimuli presented one at a time into one of several categories [51, 52]. Corrective feedback is typically given after each category response, helping the participants learn to associate each stimulus with their respective categories. Some category learning studies investigate how participants alter their information access behaviors over time as they learn to be more efficient at the task. To do this, researchers sometimes use eye tracking as a tool to record overt attention, allowing researchers to identify what a participant is looking at over the course of the experiment (e.g., Blair et al. [53], Kruschke et al. [54], Rehder and Hoffman [55]). Overall, the category learning task is a sweep through the cognitive system, tapping into information access, attention, rule-use, decision-making, learning, and memory; it is thus an excellent paradigm for comparing cognitive processes across different mediums.

Category learning tasks tend to provide reliable patterns of results. Typically, most participants learn the stimulus-category relationships within 30–60 minutes (a couple hundred trials), improving their reaction times and accuracy, and taking less time during the feedback stage to review their answer before moving on to the next trial [56]. There are common information access patterns: participants make fewer and shorter fixations to stimulus features as learning progresses [43, 55] and ignore features not relevant for classification. These strategies lead to a more optimized allocation of attention [53]. The speed and magnitude of these changes depend on the difficulty of the task: more complex categories are learned more slowly,

and more complex patterns of feature reliability, such as increasing the number of irrelevant features, leads to less optimized information access behavioral patterns [43, 53].

Lab studies of category learning have shown that increasing the cost of accessing information can contribute not only to improved category learning, but also to more optimal allocation of attention [57]. Meier and Blair [58] manipulated information access cost in a category learning task. In the high access cost condition, participants had to wait 3 seconds for the stimulus features they selected to be revealed, while those in the low-cost condition had no delay to reveal features. They found that participants in the high access cost condition more quickly favored efficient information access strategies more than those in the low-cost condition. Likewise, over a series of 4 experiments, Rajsic et al. [59] demonstrated that increasing information access costs resulted in participants learning to make more optimal decisions when choosing which features to look at. Specifically, in experiment 3, they added additional motor-costs to each fixation by requiring mouse movements to the stimulus feature to uncover the information at that location, while the no-cost group could simply look around the screen without needing to use the mouse to uncover stimulus features. They found that even this small increase in motor cost was enough to exaggerate the rate at which optimal information access patterns were adopted by participants. Besides encouraging a more memory-based strategy, access costs also affect how people plan when problem solving [60–62]. Morgan et al. [61] demonstrated that increasing information access costs made learners more resilient to interruptions during a learning task, with the highest delay involving both a mouse movement as well as a delay. While these studies tout the positive impact of information access costs, some studies outside of category learning have shown that memory-based strategies encouraged by higher access costs can result in more recall errors compared to when information is more easily accessible [63, 64]. As such, it seems that care must be taken to ensure that delays in access to task-relevant information are constructed in such a way so as to maximize performance, while not being so interruptive as to negatively impact learning outcomes.

Turning back to the current work, despite growing enthusiasm regarding the use of VR in a wide variety of industry and academic applications, much is still unknown about the degree to which prior models of cognition will apply within immersive VR environments, and when/ how the use of VR will actually impact learning behaviors and information access patterns. The current research project seeks to establish whether the predictions of previous researchers have any traction in the context of a category learning experimental paradigm. If findings from previous studies do not replicate in a VR version of this experimental task, then models of attention and learning acquisition may have to be revised or appended to account for how the difference in medium impacts learning. Alternatively, if the findings of previous research do replicate well in an analogous virtual reality task, then pre-existing models of information access behaviors and learning outcomes within the category learning domain should still remain predictive of participant performance within VR based research, and can still be relied on.

Working within the framework of the category learning paradigm, the present study compared a 2D eye tracking experiment to analogous 3D desktop-based and immersive VR implementations of a category learning task. We used a relatively straightforward category structure with two relevant and one irrelevant visual feature which is displayed in Fig 1. The two relevant features together predict which category the stimulus belongs to, while the third, irrelevant feature does not give the participant any useful information about which category this particular stimulus belongs to. This category structure is sufficiently simple enough to allow participants to master it within the experiment run-time, and the irrelevant information present in each stimulus gives participants a reason to optimize their information access behaviors to only consider information that actually helps with task performance [43, 58]. The category structure

**Fig 1. Example stimulus features and category structure.** Each of the four categories can be uniquely determined from the value of features 1 and 2, while feature 3 has no bearing on the category. Optimal information sampling is thus viewing features 1 and 2, and ignoring feature 3.

is mapped onto 3 versions of a category learning task, each differing only by modality of presentation. A 2D version of the task largely emulates the procedures of typical category learning research done previously, while a 3D version of the stimulus is presented on either a desktop computer screen, or in an immersive VR headset. Data on learning outcomes and information access behavior typically used in category learning research are then collected within each of these 3 versions of the task and are used to compare whether or not performance and information access behaviors adopted in VR during learning follow the same kinds of trends typically observed in prior category learning research.

## Methods

### Participants

The participants were 179 undergraduates who were recruited from Simon Fraser University in Canada. They received course credit for their participation. All participants had normal or corrected to normal vision. This study was approved by the Simon Fraser University Office of Research Ethics and was deemed to be minimal risk (Study Number: 2018-s0444). All participants provided written informed consent and those in the VR condition were told about the possibility of motion sickness. Each participant was given instruction on how to withdraw from the experiment if they experienced any discomfort.

### Conditions and outcome variables

Participants were randomly assigned to one of three conditions: a 3D condition where stimuli were viewed on a standard Windows 10 desktop system with a 22 inch color monitor and a standard gamepad controller was used to manipulate the stimuli to reveal features and select

categories (n = 56), a VR condition where participants viewed stimuli using an HTC Vive headset and used HTC Vive hand controllers to manipulate the stimuli to reveal features and select categories (n = 58), and an eye tracking condition (2D) where stimuli were presented on a standard Windows 10 desktop with a 22 inch screen using a Tobii X120 eye tracker to identify which features were being viewed, and a standard gamepad controller to select categories (n = 65). In accordance with the prediction that higher information access costs would impact learning-related information access behaviors, the conditions could also be described as high (VR), medium (3D), and low (2D) cost conditions.

To measure the degree to which learning outcomes and information access behaviors are impacted by modality, a number of variables which co-align with measures typically gathered during category learning experiments [43, 49, 55] were used. To capture each modality's impact on learning outcomes in a category learning experiment, we examine both accuracy and response times to assess both proficiency and efficiency in task performance. Patterns of information access behaviors were also of interest, and so we also examine time spent on each stimulus feature, the number of times individual features are examined, as well as the degree to which these fixations are optimized to prioritize relevant information over irrelevant information. Lastly, we also record the amount of time spent examining trial feedback to partially capture the degree to which participants make use of feedback when learning to categorize stimuli.

## Stimuli

In each condition, participants learned to categorize stimuli with three binary-valued features into four categories (see Fig 1). Two of these features were useful to determine the category of the stimulus, while the third was irrelevant to categorization. In all conditions, stimuli were simple geometric figures of different colors. Location and relevance of each symbol remained constant across the experiment for each participant but were counterbalanced across participants to reduce any potential bias by location or color. Cube configurations were matched across conditions so that each of the VR, 3D, and 2D conditions contained approximately the same location and relevance combinations.

As can be seen in Fig 2, stimuli for the 3D condition and the VR condition resembled cubes with indentations in the sides which contain the features from Fig 1. Because of the indentations, participants could only see one symbol at a time, no matter how they rotated the cube. To maintain the same three symbol category structure, symbols were mapped to axes of the cube rather than sides. In other words, the same symbol was visible on the two opposing sides

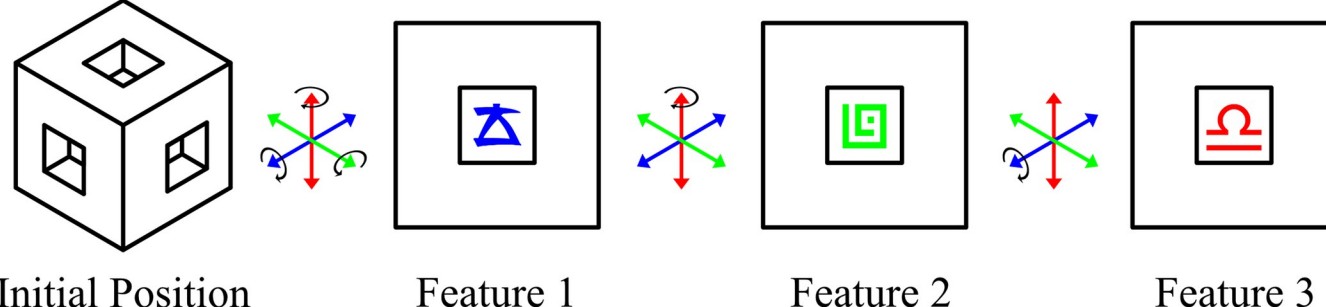

Initial Position  Feature 1  Feature 2  Feature 3

**Fig 2. A graphical representation of a stimulus cube.** The stimulus cube initially spawns in a position where none of the three features can be seen. By rotating the cube each of the three features can be seen in turn. The wells prevent more than one feature from being visible at a time. Note that opposing sides of the cube display the same feature.

for each axis. The cube was locked into a static position and could only be rotated—it could not be picked up or otherwise manipulated. In the VR condition, the cube was rotated with physical movement of the HTC Vive hand controllers, while in the 3D condition, the direction buttons on a standard gamepad controller rotated the cube. In the 2D condition, a flat 2D triangular object was used, with each stimulus feature being positioned at one of the three points of the triangle.

## Procedures

In the 3D and 2D conditions, participants were seated at a computer in an enclosed testing booth. Both conditions used handheld gamepad controllers (Logitech F310) with layouts roughly similar to the standard PlayStation DualShock controller series. Left and right shoulder buttons plus the triggers on the controller were used to select responses during each trial, and in the 3D conditions, the directional pad was used to rotate the stimulus cube. The VR condition of the experiment used an HTC Vive headset and HTC Vive hand controllers. These hand controllers were used to interact with the digital environment. In this condition participants were seated in an open testing area and were not required to move, aside from rotating the stimulus cube by colliding their hand with the stimulus cube to push it in the desired direction, and selected responses by reaching out to the desired choice-button and pulling the trigger on the controller used to touch the choice-button. Images of the controllers used in each condition can be seen in Fig 3 (adapted from [65, 66]).

After signing an informed consent form, the start of the experiment featured a tutorial which presented each participant with a short series of practice trials and a chance to ask the experimenter for guidance if a participant had trouble understanding the controls as described in the previous paragraph.

Each trial began when the participant pressed the "next trial" button which was always present on screen, either by pressing the appropriate gamepad button (for both the 3D and 2D conditions), or by positioning one of the HTC Vive hand controllers and clicking the trigger button in the VR condition. When this button was pressed, the stimulus appeared, and participants were allowed unlimited time to examine the stimulus. In the VR and 3D conditions, the stimulus features were revealed by rotating the cube to desired side, while in the 2D condition,

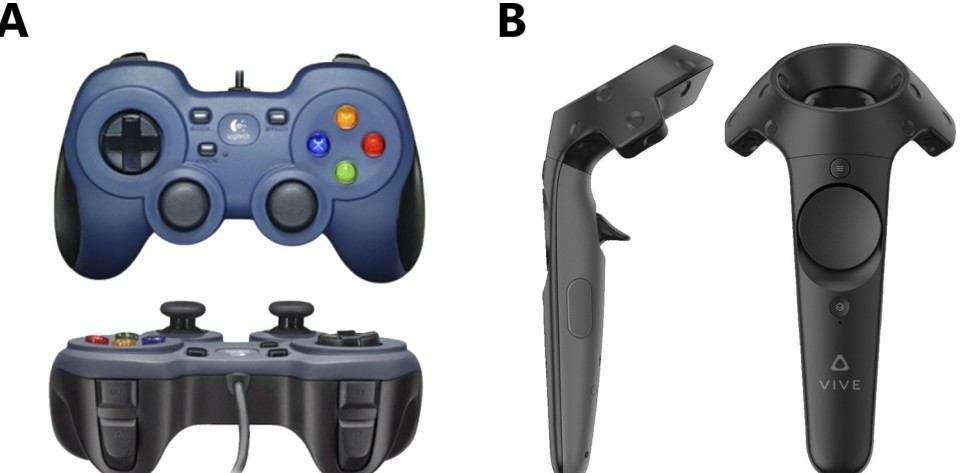

**Fig 3. The controllers used to make responses in each condition.** (A) Image of the Logitech F310 controller used in the 2D and 3D conditions (B) Image of the HTC Vive hand controllers used in the VR condition.

**Fig 4. The participants' view in the VR, 3D, and 2D conditions.** (A) An example screenshot of the view for participants in the VR condition. Specifically, this showcases the participant's view at the beginning of each trial. No stimulus features are visible. (B) An example screenshot from the 3D condition showing the participant's point of view during the feedback phase. Participant's choice shown in red, correct answer shown in green. Participant's choice lights up in green if they answered correctly. (C) The 2D stimulus as presented to the 2D group. Symbols and category structure are the same across all three conditions.

all features could be viewed by looking at the stimulus as it appeared in front of them (see Fig 4). When participants decided which category the stimulus belonged to, they reported their guess either by using the bumpers and triggers on the gamepad (2D and 3D conditions), or by reaching out to the appropriate button in VR and pressing the trigger on the Vive hand controller. After responding, they received feedback on their choice. If they chose the correct category, the correct category label for the chosen category (A, B, C, or D) turned green. If their answer was incorrect, the category label that they selected turned red (Shown in Fig 4 for the VR condition), in addition to the correct category label turning green. No time limits were used during the feedback phase, and participants could continue examining the stimulus as desired until they were ready to continue to the next trial. Once they were ready, they pressed the next trial button and were immediately sent to the next trial. The experiment ended upon completion of 240 trials or after 30 minutes, whichever came first.

## Calculation of dependent measures

**Learning outcomes.** Firstly, to capture learning outcomes, Accuracy and Response Time were used. Accuracy is defined as the proportion of correct responses made during each trial block. As in prior research, participants must reach a criterion peak performance of 24 correct trials in a row to be included in the analysis. By limiting the analysis to these participants, this accuracy measure provides information on the swiftness with which any particular group reaches that peak, as well as any baseline improvements offered by any given modality from the beginning of the experiment.

Response Time refers to the average time taken to make a categorization response after the stimulus has appeared.

**Information access behaviors.** Information access behaviors are abundant in any learning task. Of interest, we examine attentional optimization, time spent on feedback, and the average number and duration of individual fixations made prior to each category choice.

Attentional Optimization reflects the degree to which a participant emphasizes sampling task-relevant information during each trial. The measure is the difference in average time spent fixating relevant and irrelevant features over the total time spent fixating stimulus features. The measure ranges from -1 to 1. A participant spending all their time on irrelevant features would have an Optimization of -1, and all their time on relevant features would have an Optimization score of 1. If the participants spend equal time on all features, fixating the two relevant features for 1 second, and the irrelevant feature also for 1 second, they will have an Optimization score of 0, as the mean time spent per relevant feature and irrelevant feature was equivalent. As with accuracy, the trial-by-trial score was then averaged across trial blocks.

Following the category choice, Feedback Duration is captured to examine how long participants in each condition spend examining the feedback received after each trial before choosing to move on to the next trial.

To more closely examine how attention-related information access behaviors change over time, fixation durations and fixation counts were captured in all conditions to measure how long and how often individual stimulus features were fixated on during each trial block. For the 2D condition, these fixations were recorded by our eye trackers, and in the 3D and VR conditions, these were recorded using the formulas described in the preceding section.

**Detecting fixations.** The symbols used in the 2D condition were identical to the VR and 3D conditions except that all symbols were visible to the participant at once. Fixations in the 2D condition were detected by analyzing gaze data recorded by a Tobii X120 eye tracker placed below the computer screen with a sampling rate of 60Hz, and a spatial resolution of 0.5˚. Participants were positioned 60cm away from the device. Fixations were extracted by mapping gaze to the known location of the symbols using the method described by Salvucci and Goldberg [67] with a temporal threshold of 75ms, and a spatial threshold of 1.9˚. The eye tracker was calibrated for each participant.

As a parallel to eye fixations used in standard eye tracking category learning research, we defined fixations in the 3D and VR conditions based on which stimulus feature was visible to the participant at any one time. Following previous work from our lab [44], we focus on information access behaviors rather than pure attentional allocation. Bringing a symbol into view by manipulating the cube has the same high-level function of making it available to process as making an eye movement or shifting one's viewscreen across the map in a game of StarCraft.

Properties of the Unity environment used to program the experimental interface gave us enough information to detect whether a feature was visible at any given time. When symbols came in and out of view, we logged which symbol was visible and for how long. It was not possible for participants to view more than one symbol at a time. To detect when symbols were in view we used the angle of the cube faces, as shown in Fig 5. In the Unity engine, the main camera represents the view of the player, and each face of a cube is identified by an axis number. The angle of each axis to the viewing plane of the main camera can be calculated using the following C# code:

```
angle = (int) Math.Abs(Math.Round(Vector3.Angle (player.transform.forward, transform.
    right)) - 90.0)
```

Where *player.transform.forward* represents the plane of the main camera and *transform. right* represents one of the axes of the cube (the other axes are *transform.forward* and *transform.up).*

The raw angle measurement provided by Unity ranges from 0 to 180 degrees. To account for instances where the far side of the cube, which had the same symbol, was visible to the participant, the calculated angle was normalized to range from 0 to 90 degrees. At 0 degrees the symbol is completely invisible to the participant and at 90 degrees the symbol is directly in view. Testing showed that the minimum viewing angle for a symbol to be visible was 56 degrees. The calculated angles for each axis were compared to identify the axis with the greatest angle relative to the camera (i.e., the calculated angle closest to 90 degrees). If that axis was above the minimum threshold of visibility the participant was deemed to be viewing the corresponding symbol. Only one axis, and therefore, only one symbol could be viewed at any given time. The 3D condition used the same angle calculation as the VR condition, but with the computer screen representing the main camera plane instead. By calculating the viewing angle relative to the position of the camera view, only instances where the player camera is actually facing and mainly centered on a particular stimulus feature are captured, and so this approach

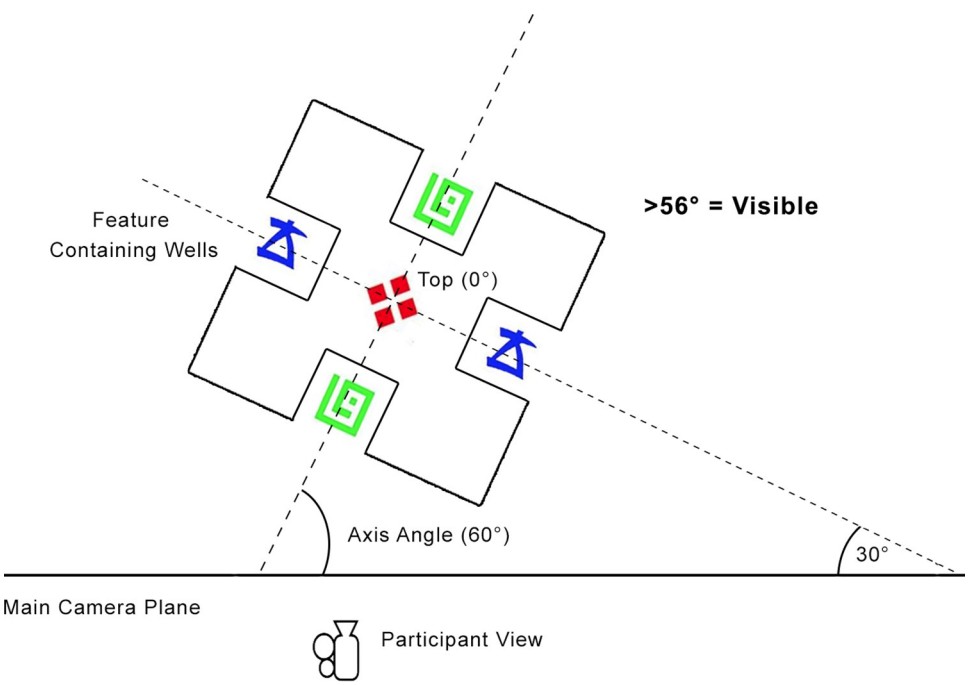

**Fig 5. The axis angle is calculated to reveal which feature is in view.** The participant rotates the cube about a central point. As the cube is rotated relative to the position of the participant view, features are exposed inside the wells. As the cube is rotated, the axis angle changes. This can be used to determine when a feature is visible to the viewer. At 56 degrees of rotation, the feature begins to be viewable to the participant.

also detects when participants are looking away at the response buttons or other irrelevant objects in the environment.

A JavaScript Object Notation (JSON) encoded data file, shared by all experimental conditions, was used to map symbols to axes. This data file contained an exhaustive enumeration of every cube/category mapping repeated for each possible cube orientation. Within each condition, individual participants viewed cubes with unique orientations and category mappings which were randomly selected from the data file.

## Exclusions

Participants who failed to meet the required learning criteria of 24 correct responses in a row were excluded from the final analysis. Among the VR group, 32 non-learners were excluded. Of these 32 non-learners, 6 dropped out after reporting physical discomfort using the VR system. There were 30 non-learners excluded from the 3D group. In our 2D sample one participant dropped out after reporting dizziness, and an equipment failure resulted in only 8 participants having complete data sets which also met the criterion point. Despite the missing data, close inspection of the data in comparison to another category learning experiment using the same structure [56] yielded no evidence that these data came from a different population distribution for any of the variables of interest. Because of the similarities between our eye tracking sample and other category learning experiments, data excluded due to equipment failure was deemed missing at random, ruling out this exclusion as a source of bias in the models.

## Results

We compared our three conditions on measures of learning outcomes and information access behaviors. We used R [68] with the lme4 package by Bates, Maechler and Bolker [69] to

perform linear mixed effects analyses of the relationship between the condition and each of the measures used. The fixed effects used in each model were the trial bin (bins 1–10) and the condition (2D, 3D, and VR). Interaction effects between bin and condition were also considered in each model. Subject ID was used as a random intercept. The largest model for each dependent variable followed the formula: DV ~ Bin + Bin$^2$ + Condition + Bin:Condition + Bin$^2$:Condition. All significance tests were done by performing a likelihood ratio test comparing models where the factor is present to ones where the factor is absent. Using this approach, we are able to compare both the overall differences between groups through main effects, as well as differences in the rates of change across experiment bins via the interaction effect in each model.

All variables of interest were found to have significant changes over the course of the experiment, indicating an effect of practice for all three groups. Bin squared (a quadratic effect) was added to each model to better describe the curvilinear nature of each group's learning curve. As expected, a significant quadratic effect on bin was found for all variables of interest: accuracy ($\chi^2 = 218.21$, $p<0.001$), optimization ($\chi^2 = 84.65$, $p<0.001$), response time ($\chi^2 = 130.32$, $p<0.001$), feedback duration ($\chi^2 = 131.48$, $p<0.001$), fixation duration ($\chi^2 = 49.01$, $p<0.001$), and fixation count ($\chi^2 = 77.32$, $p<0.001$). This indicates that all groups improved with practice, while the following analyses focus on the individual impact of each condition in addition to practice effects for each variable of interest.

### Learning outcomes

As can be seen in Fig 6A, no differences were found between the conditions in terms of baseline improvements to accuracy ($\chi^2 = 4.043$, $p = 0.133$). There were also no differences found in the interaction effect ($\chi^2 = 9.012$, $p = 0.173$). Everyone who reached the criterion point generally learned the category structure in roughly the same amount of time and improved at approximately the same rate.

Response time data for all conditions are shown in Fig 6A, and clear differences can be seen in the different conditions. The VR group and the 3D group were similar to one another; the

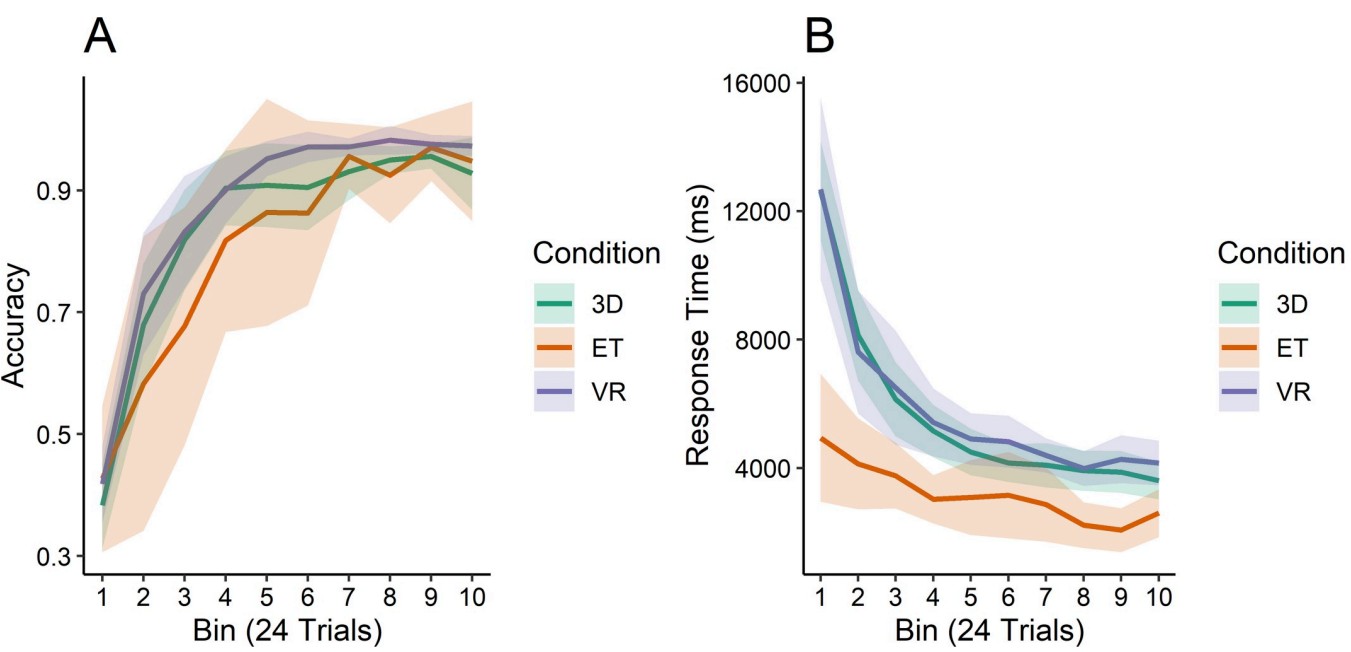

**Fig 6. Accuracy and response time results.** (A) Accuracy as a proportion and (B) response times across the 10 bins, measured in milliseconds, by condition.

participants in the 2D group generally spent much less time on each trial before making a choice (Fig 6A). Our analysis confirms these differences: we found a significant effect of the condition ($\chi^2$ = 12.339, $p$ = 0.002) with a significant interaction as well ($\chi^2$ = 40.32, $p$<0.001), but only the 2D condition stood out as different—faster, and with a lower peak. Between the VR and 3D groups, no difference was found in terms of response time.

### Information access behaviors

As for information sampling and the Optimization measure, while we found no main effect of condition ($\chi^2$ = 0.734, $p$ = 0.693), there was a significant interaction between bin and condition ($\chi^2$ = 33.489, $p$<0.001). As seen in Fig 7A, the participants in the VR and 3D appear to optimize their information sampling faster than their 2D counterparts, though the 2D condition achieves greater Optimization by the end of the experiment.

When receiving feedback on their choice, there were no differences in how long each of the groups spent considering feedback on their choices (see Fig 7B). Statistically, there was no effect detected based on condition ($\chi^2$ = 3.818, $p$ = 0.148), nor was there a significant effect for the interaction ($\chi^2$ = 5.048, $p$ = 0.282). Despite the different motor requirements for each condition, there were no discernable differences in the amount of time any group spent to review the correct categorization for each trial.

Response times conflate the number and duration of fixations–as well as any time spent not fixating features–so we plot fixation count and fixation duration separately to get an idea of how feature viewing might be different across conditions (see Fig 8A and 8B). The effect of condition on the number of fixations per trial was significant ($\chi^2$ = 13.938, $p$<0.001), with a significant interaction as well ($\chi^2$ = 15.954, $p$ = 0.003). As can be seen in Fig 8A, Participants in the 2D condition started with fewer fixations on average and maintained a flatter curve. People in the 3D condition started with more fixations than the eye tracker participants, but eventually reached the same lower asymptote as the 2D group. Lastly, the VR group started with the most fixations and did not reach the lower fixation counts of the other two groups.

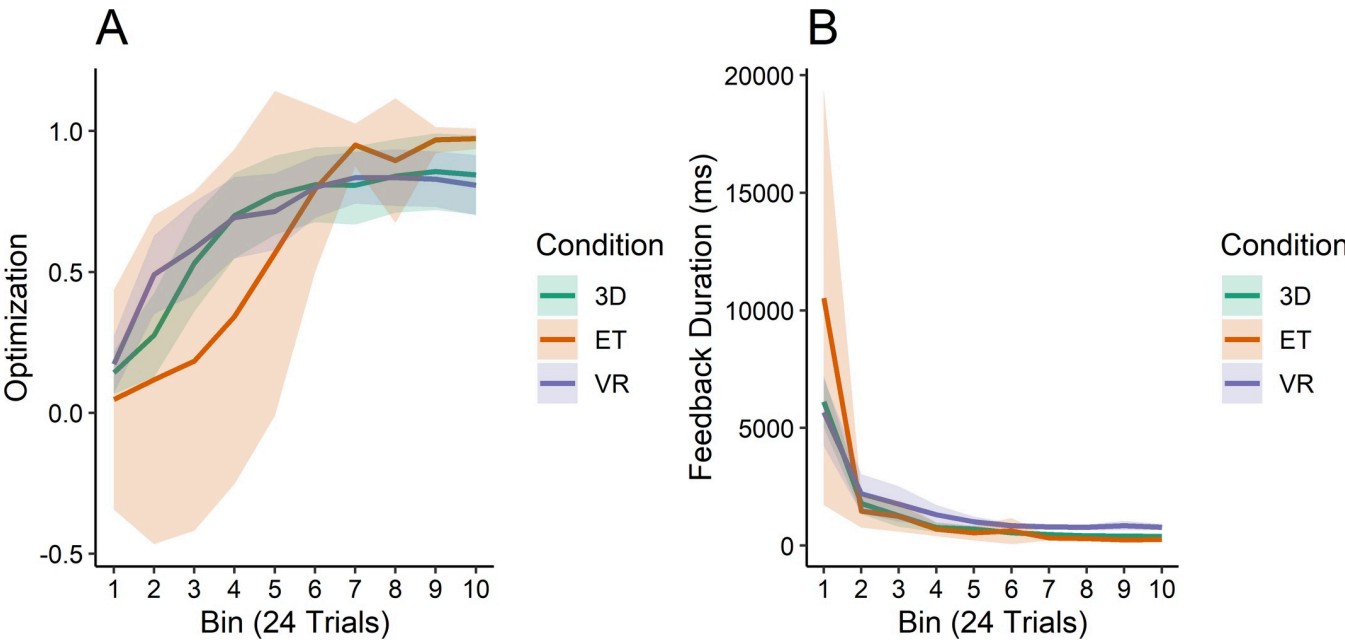

**Fig 7. Attentional optimization and feedback duration by condition.** (A) Information access optimization ranging from -1 to 1 (see text for calculation) and (B) time spent looking at feedback in between trials across 10 bins, measured in milliseconds, by condition.

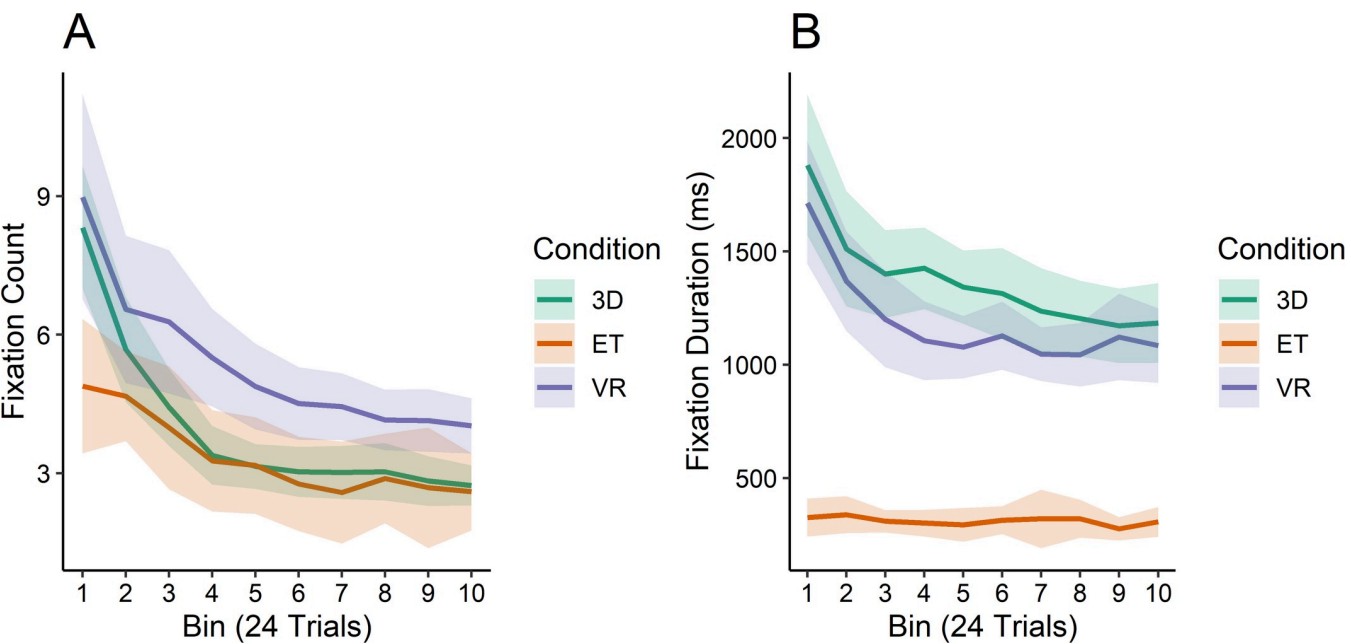

**Fig 8. Fixation counts and average fixation durations.** (A) Number of fixations per trial averaged for each bin, by condition. (B) Average fixation length in milliseconds for each bin, by condition.

The fixation durations were shorter for participants in the 2D condition than for participants in the 3D and VR conditions, which were similar to one another (see Fig 8B). Our models showed that the effect of condition on fixation durations was significant ($\chi^2 = 39.148$, $p<0.001$), as well as the interaction effect ($\chi^2 = 33.827$, $p<0.001$). The 3D and VR conditions were relatively close in terms of time spent on each side of the stimulus cube, yet participants in the 2D condition were far quicker than the other two, with an approximately flat line across the ten bins indicating that these people generally spent the same amount of time on each feature across the duration of the experiment.

## Discussion

Virtual Reality offers unique opportunities and challenges to both researchers of cognition, and designers of VR tools and environments. The unique immersion and interaction methods characteristic of VR have potential to impact both learning outcomes and information access behaviors by increasing immersion and through the costs, in time and energy, of accessing information. The present study compared learning outcomes and information access behaviors on a category learning task between three groups: one presented with 3D stimuli while immersed in the HTC Vive VR system, another presented with the same 3D stimuli while using a flat-screen desktop computer, and third presented with a 2D projection of the stimuli on a desktop computer while their eye movements were tracked. Our aim was to assess the impact of immersion and access costs on learning outcomes and information access behaviors, to better understand the impact of VR on cognition.

We found clear evidence of strong similarities across all three task implementations. First, all the expected qualitative trends for both learning outcomes and information access behaviors were obtained in all conditions. Over the course of the experiment, accuracy and optimal sampling of feature information both increased in all conditions. The number of fixations per trial, the duration of those fixations, and the overall response times for both the response and

feedback phases of each trial decreased in all conditions. This finding is in line with the medium-agnostic nature of information sampling observed in McColeman et al. [44] and suggests that findings and insights from McColeman et al. [43] on learning outcomes and information access behaviors in category learning should map cleanly onto VR and 3D implementations of these cognitive tasks. Overall, then, the general patterns in learning outcomes and information access behaviors found in extant category learning research replicate well in both 3D and VR implementations.

With respect to learning outcomes, we found no evidence that participants experienced different learning curves in terms of accuracy across the three versions or the task. This stands in contrast to previous reported findings both in terms of the alleged learning benefits of VR, and in terms of performance gains facilitated by increased information access costs [57]. This finding is in line with the numerous neutral-positive findings reported in Angel-Urdinola [14] and offers further evidence that switching mediums is not sufficient to improve learning outcomes. However, individual differences have also been found to have impacts on the degree to which VR is capable of having a positive impact on learning outcomes [70–72]. In studies which captured more information about the individual differences between participants, the positive impacts of VR were only able to outpace the learning outcomes of other instantiations when mediating differences between participants were taken into account. For instance, Sun, Wu, and Cai [72] found that learners with low spatial ability benefited the most from VR-based learning, while students with high spatial ability did no better in VR than with paper-based learning. It is possible, then, that there are subtler mediating variables influencing learning, but our findings stand with those reviewed in the introduction as evidence against the idea that VR grants a broadly applicable learning advantage.

While there were strong similarities across implementations of the task in terms of learning outcomes and patterns information access behaviors, there were also differences large and small. By far the most salient is that overall, the 3D and VR versions of the task yielded much slower response times. One might be tempted to explain difference in learning outcomes because accessing the information by rotating the cube (in 3D and VR) is a slower process, but participants also had longer Fixation Durations in these conditions, indicating that they spent longer periods of time examining each feature, and so examining this variable as a learning outcome without also considering information access behaviors cannot fully explain our results. Information access behaviors also differed in other ways. Another difference between conditions was that the VR condition had a higher number of fixations than the other two conditions. This may be due to the novelty of VR [73]. Anecdotally, many participants reported never having tried VR before participating in our study. The novelty of moving one's arm in the real world and having that translate to a virtual experience is quite appealing. Because rotating the cube was the only interactive experience within the experiment, this may have encouraged an increase in fixations as the generally pleasing nature of interactions with the cube in VR may have led participants to enjoy the experiment enough to not wish to rush through the trials. Indeed, Jensen and Konradsen [15] and Farra, Smith, & Ulrich [22] contend that the enthusiasm of participants in VR-based learning tasks may be partly driven by the duration of the tasks involved, as most studies only analyzed learning outcomes after a single learning session. It is possible that, with more exposure to the interface, the novelty effect would wear off and participants' engagement and any benefits that come with it, might decrease to more "normal" levels.

Optimization of information access behaviors was not equivalent across conditions, and the VR and 3D conditions, while exhibiting no overall differences, had a different learning trajectory from the 2D group, which showed less optimization of information access behaviors initially but greater Optimization for the 2D group by the end. This is partly consistent with

our prediction that higher access costs should lead to more optimal information sampling initially but warrants some caution. First, there is no real advantage for VR over the lesser cost of the 3D condition, which goes against the prediction that higher information access costs, moving an entire arm vs. pressing buttons with a thumb, would lead to higher optimization. Secondly, the 2D condition has higher levels of Optimization by the end, which was not predicted, as this group, being the lowest cost condition, was predicted to have the worst Optimization. These differences are difficult to explain as the possibility that in VR looking around is more fun—which is consistent with our findings of higher fixation counts in VR—leading to lower overall optimization seems at odds with the 3D condition results of a smaller increase in fixation counts that fades by the end. It is unclear what elicits these differences, and more research is needed, at the very least to replicate this result.

Overall, the change in access cost did not seem to have a strong, orderly effect in our data. Only once did the conditions differentiate themselves in order of access costs 2D < 3D < VR and that was for fixation count, and in the opposite of the predicted direction. Why was our access cost manipulation ineffective, in contrast to prior research? One possibility was that it was simply not strong enough. Another possible explanation might be that Wood et al. [57] used delays that were obvious to the participant and fixed in time; no matter how quickly the participant moved the mouse, they would still need to wait before the feature was revealed. In contrast, our VR condition did not actively impede participants' progress; delays happened simply because it takes longer to move an arm than it does to move an eye. In the 3D condition, rotation was slow by comparison, but still continuously controlled by the participant's use of the standard gamepad controller. It is possible that the perception of an access cost was not present in these cases due to the autonomy granted to the participants who could shorten delays by being more efficient with their respective user interfaces. The literature on access cost is diverse both in the conceptualization of access cost and in the cognitive processes that are engaged to study its effects. Access cost has no agreed-upon definition or operationalization; potential costs exist wherever time or effort may be expended. Temporal costs are quite variable; they can be the natural consequence of another access cost such as the need to move to some location or perform some task to access information [60, 61, 63, 64], or they can occur in isolation from other costs, such as timed masking of task-relevant information [57, 60, 61]. Effort costs can come from requirements to move somewhere in physical space [63, 64], or from an action that must be performed within a computer interface [57, 60, 61]. VR interfaces come with their own unique and varied costs, such as unfamiliar button layouts on controllers [74], novel VR versions of tasks from familiar experimental paradigms [37, 41], increased complexity/ecological validity of VR-based tasks [41], and increased cognitive load that can be induced by the physical differences between real vision and VR displays [16]. Our findings show that not all costs have impacts on learning outcomes and information access behaviors, but more work is needed to predict when and in what way, access costs will affect performance.

It is worth noting that some participants in the VR condition were excluded not simply because they were non-learners, but because they withdrew from the study citing fatigue, mild headaches or dizziness. This experiment lasted less than one hour, and so the increased rate of attrition in this group, whether it be cybersickness or other fatigue related conditions, poses a cost that must be set against other potential benefits of using these systems.

## Conclusion

The findings of the present work have implications for both researchers and designers building VR environments. For the researcher, because VR headsets are fixed to the user, they allow for

very precise control of stimulus presentation. For example, as the screen is kept in a fixed position at a constant distance from the eyes, researchers may ensure that a stimulus subtends an exact number of degrees of visual angle, regardless of participants' head movements. As the general patterns of learning outcomes and information access behaviors were found, in the current work, to largely predict performance and behavioral patterns in VR (and 3D), researchers seeking to take advantage of the affordances of this medium in their experimental design should feel comfortable doing so. Likewise, designers of VR experiences should be able to leverage research that has come before to inform their expectations of user behaviors, even if that research was not conducted directly in their target platform. Our work suggests that the base of knowledge and design principles which researchers and designers have built for 3D desktop environments should be applicable if they transition to virtual reality, and the converse applies. Design principles and useful practices that are backed by research will likely carry between domains; this reduces the need to run repeated studies and allows designers to rely on existing systems. More research is clearly needed to understand how access costs influence information access behaviors. Alongside studies like Soret et al. [38], Li et al. [4], and Eichert et al. [46], our research, being the first to combine the methods of the category learning paradigm with immersive VR technology, provides a foundation for future studies aiming to use VR in the study of cognitive phenomena.

## Acknowledgments

Special thanks and acknowledgements to the following people for helping with this project: Ximin Zhang, Ishan Sood, Thomas Nakagawa, Emma Hughson, Alex Volkanov, Atticus Shi, and Tyrus Tracey.

## Author Contributions

**Conceptualization:** Robin Colin Alexander Barrett, Justin William O'Camb, Ruilin Zhang, Mark Randall Blair.

**Data curation:** Robin Colin Alexander Barrett, Justin William O'Camb, Cal Woodruff, Scott Marcus Harrison, Katerina Dolguikh, Mark Randall Blair.

**Formal analysis:** Robin Colin Alexander Barrett, Justin William O'Camb.

**Investigation:** Robin Colin Alexander Barrett, Justin William O'Camb, Katerina Dolguikh, Ruilin Zhang, Mark Randall Blair.

**Methodology:** Mark Randall Blair.

**Project administration:** Robin Colin Alexander Barrett, Rollin Poe, Justin William O'Camb, Mark Randall Blair.

**Resources:** Robin Colin Alexander Barrett, Mark Randall Blair.

**Software:** Robin Colin Alexander Barrett, Rollin Poe, Justin William O'Camb, Cal Woodruff, Scott Marcus Harrison, Katerina Dolguikh, Christine Chuong, Ruilin Zhang.

**Supervision:** Robin Colin Alexander Barrett, Justin William O'Camb, Katerina Dolguikh, Mark Randall Blair.

**Validation:** Robin Colin Alexander Barrett, Justin William O'Camb, Cal Woodruff, Scott Marcus Harrison, Katerina Dolguikh, Rohan Ben Joseph, Mark Randall Blair.

**Visualization:** Robin Colin Alexander Barrett, Justin William O'Camb, Cal Woodruff, Ruilin Zhang.

**Writing – original draft:** Robin Colin Alexander Barrett, Rollin Poe, Justin William O'Camb, Cal Woodruff, Scott Marcus Harrison, Christine Chuong, Amanda Dawn Klassen, Ruilin Zhang, Rohan Ben Joseph, Mark Randall Blair.

**Writing – review & editing:** Robin Colin Alexander Barrett, Rollin Poe, Justin William O'Camb, Cal Woodruff, Scott Marcus Harrison, Katerina Dolguikh, Christine Chuong, Amanda Dawn Klassen, Ruilin Zhang, Rohan Ben Joseph, Mark Randall Blair.

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
