## [Decision Letter · Decision Letter 0]

22 Apr 2022

PONE-D-22-02993Comparing Virtual Reality, Desktop-Based 3D, and 2D Versions of a Category Learning ExperimentPLOS ONE

Dear Dr. Barrett,

Thank you for submitting your manuscript to PLOS ONE. After careful consideration, we feel that it has merit but does not fully meet PLOS ONE’s publication criteria as it currently stands. Therefore, we invite you to submit a revised version of the manuscript that addresses the points raised during the review process.

We look forward to receiving your revised manuscript.

Kind regards,

Hong-jin Sun, Ph.D.

Academic Editor

PLOS ONE

Journal Requirements:

Reviewers' comments:

Reviewer's Responses to Questions

**Comments to the Author**

1. Is the manuscript technically sound, and do the data support the conclusions?

Reviewer #1: Partly

Reviewer #2: Partly

2. Has the statistical analysis been performed appropriately and rigorously? 

Reviewer #1: No

Reviewer #2: N/A

3. Have the authors made all data underlying the findings in their manuscript fully available?

Reviewer #1: Yes

Reviewer #2: Yes

4. Is the manuscript presented in an intelligible fashion and written in standard English?

Reviewer #1: No

Reviewer #2: Yes

5. Review Comments to the Author

Reviewer #1: SUMMARY

Researchers used a basic category learning task to test whether a virtual reality version would be comparable to desktop 3D and 2D versions of the task. Because participants generally enjoy virtual reality tasks more than desktop tasks, ensuring that the results are parallel is important. Participants were 179 undergraduate students, although roughly half the participants’ data were not included because they did not learn the task sufficiently under the current constraints. The measures of performance (accuracy, RT), visual scanning optimization, and fixation counts and duration, all showed similar results in that the data was best described using a quadratic function. The only exception was that the 2D task differed on a couple of the measures (RT, fixation duration). Finding that the participants in the virtual reality condition performed similarly to the participants in the desktop condition means that the relatively tedious task can be replaced with the more enjoyable one.

EVALUATION

It is critically important to show that tasks in a virtual environment function equivalently to tasks in typical desktop 3D and paper-and-pencil 2D tasks. This is because a virtual environment has as much experimental control as the others but is more ecologically relevant, overcoming a common criticism of 3D and 2D tasks. Therefore, the basic premise of the paper is valid, and the results would be of interest to cognitive psychologists and neuroscientists.

However, I have several concerns about the current manuscript, as detailed below. I reserve judgment about the discussion until these are addressed.

1. The introduction should be refocused.

Although the structure of the introduction is reasonable, it obscures the information that is directly relevant to the current project. Currently, the introduction describes the studies that show similar results between virtual and desktop/paper tasks, and then those that show different results. In addition, it describes the benefits of using virtual tasks, including that people enjoy learning tasks virtually, and in some cases, the task is more realistic and therefore more relevant for learning the task. But, the tasks included in the introduction vary widely in terms of embodiment and very few use basic cognitive tasks. I suggest the following changes.

a. The introduction should focus on research that is more similar in terms of basic cognitive processes. Provide more detail about the tasks and the results. For the results, evaluate the learning process: initial task learning, learning rate across time, and asymptotic performance.

b. Make a more convincing case for using virtual tasks over traditional tasks; for example, focus on the enjoyment aspect and ecological validity (immersion). These were mentioned but should stand out more. My previous comments highlight the enjoyment argument because that may be the best one to use for a basic cognitive task. At one point the authors pointed out that a virtual task could be used in fMRI research, but one cannot move one’s head in the machine, so this does not seem to be a benefit.

c. On the flip side of the case, researchers who do not have access to virtual equipment can make the argument that enjoyment of the task (or whatever argument is made for virtual tasks) does not matter – it does not affect performance – and so they can be comfortable using the traditional tasks. This reasoning could be a further benefit for using a virtual task, or it could raise the question of why the current study is important (i.e., why bother?).

2. The tasks and procedures need more explanation.

a. I did not understand the logistics of what participants did in the tasks. In the 2D tasks I am assuming that they could see all features at a glance, but how was the task presented and how did participants respond? On a desktop computer? Using a joystick? Pencil/paper? Further, what was presented in the 3D condition? The same as in the virtual condition (Figure 4)? How is the virtual controller different than the joystick? If the virtual task used a virtual joystick, it seems that the task itself is identical and only the surrounding environment is different (i.e., looking at the stimuli on a computer in a room vs. being in a room, with the stimuli floating in front of you). Why would this difference have an impact on performance? Again, why bother with a virtual task?

b. How many trials were there? In the results section it says that there were 24 trials per bin, implying 240 trials. I am assuming that each category was presented equally, so 60 trials each. Did participants receive 10 (continuous) blocks of trials, each with 4 categories x 6 presentations = 24 trials? How were the trials randomized? This information should be in the methods section.

c. The dependent measures should be operationalized in more detail. For example, reaction time is measured as the time between the presentation of the cube-to-be-categorized and [what?].

3. The data analyses need to be more in-depth.

For each measure, there is an initial stage of getting familiar with the task (Bin 1), a learning curve (Bin 2 to asymptotic performance) and an overall difficulty of the task (asymptotic performance, or Bin 10). The learning curve and asymptotic performance should be analyzed separately. This would provide a better comparison of learning differences among the tasks, and whether they are comparable. Any discussion should focus on these more subtle differences among the groups. Here are three examples.

a. Accuracy. Asymptotic performance for all groups appears to be around .90. The VR group appears to reach asymptote at Bin 5, whereas the other groups do not reach asymptote until Bin 10. Surely the learning curve (i.e., slope) differs among the groups? What does this mean? Why would the VR group learn the task more quickly?

b. Optimization. Asymptotic performance for all groups appears to be around .80. The ET/2D group reaches optimization at Bin 4, whereas the other groups reach it at Bin 6 or 7. What does it mean that the VR and 3D optimize fixations on relevant features at the same rate?

c. Response Time. The ET/2D group is dramatically faster even in Bin 1. Any further analysis may need to use Bin 1 as a covariate. The ET/2D reaches asymptote at Bin 3 or 4, whereas the other groups reach asymptote at Bin 8 or 9 (if at all). Again, what do the differences and similarities mean in terms of the cognitive processes involved in category learning?

4. The number of non-learners is concerning.

I understand that the percentages of non-learners may be similar to previous research, but the number (nearly 50%) is still concerning. What is happening with these participants?

A criterion of 24 consecutive correct trials was set as the inclusion criterion. This means that they had to have reached asymptotic performance to be included. I understand that any model of cognitive processes should include only performance that is accurate. However, even though the non-learners may not have reached asymptotic performance within 240 trials, their learning curves are still informative. A good cognitive model should include when/where there are breakdowns in performance. If their data are too variable, use the median rather than the mean for each bin, where possible. At some point I would like to see the non-learner’s data, even if not in this manuscript.

5. The inclusion of a 2D/ET group completing a task with different stimuli is concerning.

It is unclear whether the differences in performance for the 2D group is because of the task or the stimuli. Either gather more data with the current stimuli or state that there were not enough remaining participants to analyze. Or, you could analyze the data you have. Your data are stable, with each measure in each bin averaged across 24 trials per person. When this is the case, we often only need 8-10 participants per group to find an effect.

Reviewer #2: The reviewed paper describes user testing of virtual reality (VR) in the scope of application in learning. VR was compared with 2D and 2D visualization. This topic is undoubtedly current and interesting. I have a few comments on the methodology used, especially its description.

Introduction

- Because I have a background in geospatial sciences, I know of several examples of VR applications in this area, such as teaching geography.

- Some other studies compare desktop VR, and immersive VR created using HMD (e.g., Zhao et al., 2020; https://www.tandfonline.com/doi/full/10.1080/13875868.2020.1817925).

Methods

- None of the participants were excluded due to cybersickness?

- On the one hand, I understand that equipment failures can occur during the experiment; on the other hand, it seems a bit amateurish.

- I would recommend describing the symbols that served as stimuli in the experiment. On what basis were they selected? Does their selection have any basis in literature?

- Determining the fixations for the 3D and VR conditions seems inappropriate. Even if the user had the cube turned to the given symbol, this does not mean that he did not look in the virtual environment other than this cube wall.

- HTC Vive can be combined with an eye-tracking device (HTC Vive Pro Eye headset). Why did not the authors try to take advantage of this opportunity? Eye-tracking in VR is described in detail by, for example, Ugwitz et al. (2022; https://doi.org/10.3390/app12031027).

- It follows from the above that the term "eye-tracking condition" is not appropriate. I recommend replacing the "2D condition" in all parts of the manuscript (e.g., in the abstract, it is correct).

- I recommend adding illustrative photography of the participants in all three conditions. For example, it will answer whether they sat or stood during the solved tasks.

- The terms "VR hand controller" and "standard game controller" are unclear. Better describe what kind of device it is, for example, that it was an HTC Vive controller.

Discussion

- I would discuss the experiment results in the context of information equivalence.

6. PLOS authors have the option to publish the peer review history of their article (what does this mean?). If published, this will include your full peer review and any attached files.

Reviewer #1: **Yes: **Beverly Roskos

Reviewer #2: No

---

## [Author Response · Author response to Decision Letter 0]

6 Jun 2022

Thank you to both reviewers for their constructive and insightful reviews of our manuscript. As directed in the letter we received from the journal staff, we have uploaded our specific reviewer and editor comment responses as a separate file labeled 'Response to Reviewers'. This can be found attached to this digital resubmission, and we look forward to receiving any additional feedback that may be forthcoming regarding this revised manuscript's publication. I have also copied and pasted the text from the attached file to the text field below, but feel that the readability is much improved in the attached pdf file. As well, we include a figure in our pdf showing non-learner trends and this image is not capable of being displayed in this text field. As such, the reader is strongly encouraged to review the attached pdf to see the full extent of our response to reviewer comments.

Responses to Reviewers

EVALUATION

It is critically important to show that tasks in a virtual environment function equivalently to tasks 

in typical desktop 3D and paper-and-pencil 2D tasks. This is because a virtual environment has 

as much experimental control as the others but is more ecologically relevant, overcoming a 

common criticism of 3D and 2D tasks. Therefore, the basic premise of the paper is valid, and 

the results would be of interest to cognitive psychologists and neuroscientists.

However, I have several concerns about the current manuscript, as detailed below. I reserve 

judgment about the discussion until these are addressed.

1. The introduction should be refocused.

Although the structure of the introduction is reasonable, it obscures the information that is 

directly relevant to the current project. Currently, the introduction describes the studies that 

show similar results between virtual and desktop/paper tasks, and then those that show 

different results. In addition, it describes the benefits of using virtual tasks, including that 

people enjoy learning tasks virtually, and in some cases, the task is more realistic and therefore 

more relevant for learning the task. But, the tasks included in the introduction vary widely in 

terms of embodiment and very few use basic cognitive tasks. I suggest the following changes.

a. The introduction should focus on research that is more similar in terms of basic cognitive 

processes. Provide more detail about the tasks and the results.

>> We have made extensive revisions to the introduction. We have condensed or cut 

many of the more peripheral citations, and have made it clearer that there has been no 

work examining category learning in VR, and thus our work makes a novel contribution 

in this regard. Despite no directly equivalent papers, we have increased the amount of 

detail on the most similar work in category learning using common computer 

presentations. 

For the results, evaluate the learning process: initial task learning, learning rate across time, 

and asymptotic performance.

>> We have included new analyses to assess these variables. They were not included 

initially because there are few detailed analyses in the existing literature, prohibiting 

comparison from our data to existing findings. 

b. Make a more convincing case for using virtual tasks over traditional tasks; for example, 

focus on the enjoyment aspect and ecological validity (immersion). These were mentioned but 

should stand out more. My previous comments highlight the enjoyment argument because that 

may be the best one to use for a basic cognitive task. 

>> In the revised introduction we make clearer that the findings of our study provide 

evidence about the similarities and differences between VR and traditional computing, 

from a learning/information access cost perspective. The research is enlightening 

regardless of the findings: if VR were better, then our study would support its use for 

learning/optimization reasons; if it was similar, the VR designers might rely on studies 

which use traditional methods with expectations their findings would also apply to VR, 

and scientists might use VR methods with the expectation that it will not markedly skew 

the results. If VR was worse or markedly different somehow, then everyone should be 

careful with generalizing across implementations. It has never been our goal to advocate 

for VR. In our revised introduction we make sure that this is clearer, and include special 

mention in the last paragraph of the introduction, clarifying the intended purpose of the 

research. 

We also take the reviewers advice and include new discussion of enjoyment and 

engagement as a possible reason for wanting to favour VR-based methods in research. 

At one point the authors pointed out that a virtual task could be used in fMRI research, but one 

cannot move one’s head in the machine, so this does not seem to be a benefit.

>> Immersive VR is being used in a wide variety of research applications, including the 

neurosciences, which has, as the reviewer notes, significant constraints. Our mention of 

this research here was only intended to support the idea that despite differences in user 

interfaces, VR had been found by these researchers to still activate similar cognitive 

processes to those activated during identical tasks in other modalities. 

c. On the flip side of the case, researchers who do not have access to virtual equipment can 

make the argument that enjoyment of the task (or whatever argument is made for virtual tasks) 

does not matter – it does not affect performance – and so they can be comfortable using the 

traditional tasks. This reasoning could be a further benefit for using a virtual task, or it could 

raise the question of why the current study is important (i.e., why bother?).

>> As described above, it was not our research goal to establish that VR is better. Our 

contribution provides evidence of similarities and differences across implementations of 

a category learning experiment, the understanding of which will be useful for both 

researchers and designers. Our introduction was rewritten to make this more clear, 

highlighting the implications of either finding 

2. The tasks and procedures need more explanation.

a. I did not understand the logistics of what participants did in the tasks. In the 2D tasks I am 

assuming that they could see all features at a glance, but how was the task presented and how 

did participants respond? On a desktop computer? Using a joystick? Pencil/paper? Further, 

what was presented in the 3D condition? The same as in the virtual condition (Figure 4)? How 

is the virtual controller different than the joystick? If the virtual task used a virtual joystick, it 

seems that the task itself is identical and only the surrounding environment is different (i.e., 

looking at the stimuli on a computer in a room vs. being in a room, with the stimuli floating in 

front of you). Why would this difference have an impact on performance? Again, why bother 

with a virtual task?

>> We have included additional description that addresses these questions in the 

Method section for readers less familiar with the category learning paradigm. 

b. How many trials were there? In the results section it says that there were 24 trials per bin, 

implying 240 trials. I am assuming that each category was presented equally, so 60 trials each. 

Did participants receive 10 (continuous) blocks of trials, each with 4 categories x 6 

presentations = 24 trials? How were the trials randomized? This information should be in the 

methods section.

>> Yes it should; we are sorry for the omission. We have completely revised the Methods 

section to provide more complete information which answers these questions and 

others. 

c. The dependent measures should be operationalized in more detail. For example, reaction 

time is measured as the time between the presentation of the cube-to-be-categorized and 

[what?].

>> We have included additional description of methods, and apologize for not having 

done a more thorough job in the initial submission. 

3. The data analyses need to be more in-depth.

For each measure, there is an initial stage of getting familiar with the task (Bin 1), a learning 

curve (Bin 2 to asymptotic performance) and an overall difficulty of the task (asymptotic 

performance, or Bin 10). The learning curve and asymptotic performance should be analyzed 

separately. This would provide a better comparison of learning differences among the tasks, 

and whether they are comparable. Any discussion should focus on these more subtle 

differences among the groups. Here are three examples.

a. Accuracy. Asymptotic performance for all groups appears to be around .90. The VR group 

appears to reach asymptote at Bin 5, whereas the other groups do not reach asymptote until 

Bin 10. Surely the learning curve (i.e., slope) differs among the groups? What does this mean? 

Why would the VR group learn the task more quickly?

>> We did not initially include such analyses because they are not standard for the field, 

and so any such findings will have no statistically assessed comparison point in existing 

data. Nevertheless, we have now included additional analyses as requested, which also 

enlarge the discussion slightly as we consider the implications of these results.

b. Optimization. Asymptotic performance for all groups appears to be around .80. The ET/2D 

group reaches optimization at Bin 4, whereas the other groups reach it at Bin 6 or 7. What does 

it mean that the VR and 3D optimize fixations on relevant features at the same rate?

>> Done. We conducted similar block inclusive analyses and found interaction effects for 

many of the dependent variables, suggesting differing rates of learning and asymptotic 

performance between groups. These differences are now stated in the Results section. 

Since the data have been changed (see comment 5 response), the specific differences 

observed in these comments are no longer applicable. 

c. Response Time. The ET/2D group is dramatically faster even in Bin 1. Any further analysis 

may need to use Bin 1 as a covariate. The ET/2D reaches asymptote at Bin 3 or 4, whereas the 

other groups reach asymptote at Bin 8 or 9 (if at all). Again, what do the differences and 

similarities mean in terms of the cognitive processes involved in category learning?

>> As above (3a & b). 

4. The number of non-learners is concerning.

I understand that the percentages of non-learners may be similar to previous research, but the 

number (nearly 50%) is still concerning. What is happening with these participants?

A criterion of 24 consecutive correct trials was set as the inclusion criterion. This means that 

they had to have reached asymptotic performance to be included. I understand that any model 

of cognitive processes should include only performance that is accurate. However, even 

though the non-learners may not have reached asymptotic performance within 240 trials, their 

learning curves are still informative. A good cognitive model should include when/where there 

are breakdowns in performance. If their data are too variable, use the median rather than the 

mean for each bin, where possible. At some point I would like to see the non-learner’s data, 

even if not in this manuscript.

>> Our aim, to assess the link between VR procedures to prior 2D versions necessitated 

using standard category learning procedures. The category structure on which we based 

the current work and its associated standard procedures has been known to produce 

many non-learners, and in choosing this particular procedure, we inherited its associated 

non-learner rates. Nevertheless, we completely agree with the reviewer’s sentiment. We 

are currently doing experimental work to understand possible reasons these individual 

differences, but such an evaluation (long past due in our view) goes beyond the present 

focus on VR, nor is it exclusive to the category structure used here. 

To give the reviewer insight into the non-learner data, we include similar plots from the 

manuscript (available in the attached response to reviewers pdf file), except using only non-learners' data below. These plots show the mean and 

standard error for each bin within the separate groups. As visible in the first figure, the 

non-learners' accuracy was only moderately better than random chance (25%). The 

overall mean accuracy for all non-learners was 31.97%. The low accuracy, mostly 0 

optimization, low fixation counts, and short feedback durations show patterns of 

participants who possibly gave up at some point the experiment, or were bored and 

clicked through trials rapidly in order to finish the experiment faster. The main exception 

to 0 optimization is the ET condition. The non-learner ET data is rather sporadic due to 

the equipment failures (as indicated by the large fluctuations in standard error). All bins 

in which ET participants had perfect or near perfect optimization had an average of two 

or less fixations, low accuracy (all below 60%, most at 25%), and fast response times, 

indicating the participants likely looked at one or both of the relevant features due to 

random chance. 

5. The inclusion of a 2D/ET group completing a task with different stimuli is concerning.

It is unclear whether the differences in performance for the 2D group is because of the task or 

the stimuli. Either gather more data with the current stimuli or state that there were not enough 

remaining participants to analyze. Or, you could analyze the data you have. Your data are 

stable, with each measure in each bin averaged across 24 trials per person. When this is the 

case, we often only need 8-10 participants per group to find an effect. 

>> Following your advice, we have returned to the data collected during our initial data 

collection period, running our pre-planned analyses on the original data collected, and 

using the alternative dataset for quality assurance purposes so as to assess the 

reliability of the data given its smaller sample. Using this data does not change our 

overall conclusions as there are only minor differences between these two sets of data. 

**Reviewer #2:** 

The reviewed paper describes user testing of virtual reality (VR) in the scope of application in 

learning. VR was compared with 2D and 2D visualization. This topic is undoubtedly current and 

interesting. I have a few comments on the methodology used, especially its description.

Introduction

- Because I have a background in geospatial sciences, I know of several examples of VR 

applications in this area, such as teaching geography.

- Some other studies compare desktop VR, and immersive VR created using HMD (e.g., Zhao 

et al., 2020; ~https://www.tandfonline.com/doi/full/10.1080/13875868.2020.1817925~).

>> Thank you for sharing this study with us. There are a wide variety of studies being published in this field, and it is always nice to learn of other areas where VR has been applied.

Methods

- None of the participants were excluded due to cybersickness?

>> We have updated our exclusion criteria to specify that some of the non-learners were 

excluded for dropping out after reporting mild discomfort. 6 participants in the VR 

condition dropped out for this reason, as did 1 participant in the ET condition as well.

- On the one hand, I understand that equipment failures can occur during the experiment; on 

the other hand, it seems a bit amateurish.

>> Our recent equipment failures are due to our well-used tools, befitting our extensive 

experience in eye-tracking and category learning (e.g., Blair, Watson, Walshe & Maj, 2009; 

Blair, Watson & Meier, 2009; McColeman, Ancell & Blair, 2011; Meir & Blair 2012, Chen, 

Meier and Blair, 2013; McColeman & Blair; 2013; McColeman et al., 2014; Barnes, Blair, 

Tupper and Walshe, 2021; Dolguikh, Tracey & Blair, 2021). 

- I would recommend describing the symbols that served as stimuli in the experiment. On what 

basis were they selected? Does their selection have any basis in literature?

>> The symbols are shown in figure 1 of the manuscript. Stimuli in category learning 

often vary, for example, Shepard, Hovland and Jenkins (1961) use multiple objects; Medin 

and Schaffer (1978) use shapes of difference colours and numbers; Smith and Minda 

(1998) use cartoon bugs, and so on. Counterbalancing is important for ensuring minor 

differences in feature salience do not influence global findings, and the present work 

takes all the standard precautions. 

- Determining the fixations for the 3D and VR conditions seems inappropriate. Even if the user 

had the cube turned to the given symbol, this does not mean that he did not look in the virtual 

environment other than this cube wall.

>> The worry is unfounded. In our experiment, fixations in both the VR and 3D condition 

are determined, in part, by the participants head position (the plane of the main camera 

is linked to the headset) as well in as rotation of the cube. If participants are looking 

around at the peripheral environment and not at the cube it will not register as a fixation. 

The possibility exists, of course, that participants will ‘zone out’ and fixate but not 

cognitively process the information (Simons, 1999); however, that possibility exists 

equally in all eye-tracking work, and not just in those conditions. Finally, we also note 

that we have published other work in journal Attention, Perception and Psychophysics 

that used this terminology (McColeman et al., 2020). Consistency with this published 

work requires us to retain the current terminology. 

- HTC Vive can be combined with an eye-tracking device (HTC Vive Pro Eye headset). Why 

did not the authors try to take advantage of this opportunity? Eye-tracking in VR is described 

in detail by, for example, Ugwitz et al. (2022; ~https://doi.org/10.3390/app12031027~).

>> It can. The article you cite (Ugwitz et al., 2022) also describes in detail the limitations 

of current implementations of eye-tracking in VR. We were cognizant of these limitations 

when deciding on our equipment needs in 2018, and in the end, chose to use the 

equipment we had available to us. 

- It follows from the above that the term "eye-tracking condition" is not appropriate. I 

recommend replacing the "2D condition" in all parts of the manuscript (e.g., in the abstract, 

it is correct).

>> There must be some confusion: the eye tracking condition literally used an eye tracker - a Tobi x120. We have made significant revisions to the Method section in hopes of reducing the chance of avoiding these kinds of confusions; sorry for the trouble. 

- I recommend adding illustrative photography of the participants in all three conditions. For 

example, it will answer whether they sat or stood during the solved tasks.

>> All participants sat for the experiment per the description in the revised method 

section. Given that the paper has many figures already, we are reluctant to include more, 

without additional justification. We do not feel very strongly about this issue however, if 

the editor is in agreement, we can certainly add those images upon request. 

- The terms "VR hand controller" and "standard game controller" are unclear. Better describe 

what kind of device it is, for example, that it was an HTC Vive controller.

>> Done. 

Discussion

- I would discuss the experiment results in the context of information equivalence.

>> "Information equivalence" is not a term used in the category learning literature as far 

as we are aware, and we are not familiar with its technical use. Given that we do not 

really know what is intended or how to implement this comment, we await further 

clarification.

---

## [Decision Letter · Decision Letter 1]

27 Jul 2022

PONE-D-22-02993R1Comparing Virtual Reality, Desktop-Based 3D, and 2D Versions of a Category Learning ExperimentPLOS ONE

Dear Dr. Barrett,

Thank you for submitting your manuscript to PLOS ONE. After careful consideration, we feel that it has merit but does not fully meet PLOS ONE’s publication criteria as it currently stands. Therefore, we invite you to submit a revised version of the manuscript that addresses the points raised during the review process.

We look forward to receiving your revised manuscript.

Kind regards,

Hong-jin Sun, Ph.D.

Academic Editor

PLOS ONE

Journal Requirements:

Reviewers' comments:

Reviewer's Responses to Questions

**Comments to the Author**

1. If the authors have adequately addressed your comments raised in a previous round of review and you feel that this manuscript is now acceptable for publication, you may indicate that here to bypass the “Comments to the Author” section, enter your conflict of interest statement in the “Confidential to Editor” section, and submit your "Accept" recommendation.

Reviewer #1: (No Response)

Reviewer #2: All comments have been addressed

2. Is the manuscript technically sound, and do the data support the conclusions?

Reviewer #1: Yes

Reviewer #2: Yes

3. Has the statistical analysis been performed appropriately and rigorously? 

Reviewer #1: Yes

Reviewer #2: I Don't Know

4. Have the authors made all data underlying the findings in their manuscript fully available?

Reviewer #1: Yes

Reviewer #2: No

5. Is the manuscript presented in an intelligible fashion and written in standard English?

Reviewer #1: No

Reviewer #2: Yes

6. Review Comments to the Author

Reviewer #1: In this revision, the authors have been very responsive to the reviewers’ comments. The introduction and methods are much improved. I still have concerns about the presentation of the results though. The main suggestion I have is to reorganize the results to focus on (a) learning performance and (b) information seeking behaviors.

In the results section, I think that the overall regression model should be stated first and more concisely, stating that all DVs were analyzed this way.

DV = Bin + Condition + Bin* Condition + Bin-squared + Bin-squared*Condition

Separate the six DVs in terms of those that are measures of learning performance and those that are measures of the information seeking behaviors. Presumably, accuracy and response times are learning-related, and fixation count, fixation duration, attentional optimization, and feedback duration are related to information seeking.

A main effect of Bin (or Bin-squared) is expected with all DVs, indicating that performance improves with practice. The main question is whether there is a main effect of condition, and, more importantly, whether condition interacts with the linear and/or quadratic effects of Bin. So, the results should focus on these interactions.

Organizing the information in terms of learning performance and information-seeking behaviors should be reflected in the introduction, methods, and discussion, as well as in the results section. Be sure to use consistent terms throughout the paper for “information-seeking behaviors” – sometimes they are referred to as information access, information sampling strategies, information sampling, or attentional optimization.

Other suggestions follow.

a. Include a photo or drawing of the controllers in the three conditions.

b. What are the outcome measures used in the Hamilton et al. meta-analysis? (line 152)

c. The comparison group is missing in several places (e.g., measurable improvements over what?) (lines 153, 165, 178, 186, 226, 237, 243)

d. Add the DVs to Conditions in the Methods section (line 290); for example, “Conditions and Outcome Variables” but save how they were calculated for later in the methods section, in a section called “Calculation of Dependent Measures” (or something like that). “Detecting Fixations” would become part of this new section.

e. The paragraph describing a trial (lines 342-353) is still confusing. It should be in the form: A trial began with_____, then the stimulus appeared, and the participant manipulated the object (VR and 3D conditions) or simply looked at the stimulus (2D condition). When they decided which category the stimulus belonged to, they _____. After responding, they received feedback until they were ready for the next trial, at which time they pressed ____. This produced a arrow with the words “next trial.” When the participant pressed ____, the next trial began.

Reviewer #2: Most of my suggestions have been responded to in the manuscript and commented on in the cover letter. The structure of the paper has been improved, and the description of the methodology has been improved. I do not understand why the authors insist on naming the eye-tracking ("ET") condition. J still believe that this tested variant should be marked as "2D".

7. PLOS authors have the option to publish the peer review history of their article (what does this mean?). If published, this will include your full peer review and any attached files.

Reviewer #1: **Yes: **Beverly Roskos

Reviewer #2: No

---

## [Author Response · Author response to Decision Letter 1]

13 Aug 2022

Category-VR Aug2022 PLoS ONE Revise and Resubmit

Responses to Reviewer Comments

Reviewer #1: 

In this revision, the authors have been very responsive to the reviewers’ comments. The introduction and methods are much improved. I still have concerns about the presentation of the results though. The main suggestion I have is to reorganize the results to focus on (a) learning performance and (b) information seeking behaviors.

Response: Thanks again for your constructive criticism. Your comments have been well appreciated. We Agree with the above suggestion and have made an effort to follow through with this suggestion. Condensing the results into two primary concepts makes for a much more approachable interpretation of the results.

In the results section, I think that the overall regression model should be stated first and more concisely, stating that all DVs were analyzed this way: DV = Bin + Condition + Bin* Condition + Bin-squared + Bin-squared*Condition

Response: Done. 

Separate the six DVs in terms of those that are measures of learning performance and those that are measures of the information seeking behaviors. Presumably, accuracy and response times are learning-related, and fixation count, fixation duration, attentional optimization, and feedback duration are related to information seeking.

Response: Sub-headers have been implemented across the paper to better distinguish variables relating to learning outcomes and text relating to information access behaviors.

A main effect of Bin (or Bin-squared) is expected with all DVs, indicating that performance improves with practice. The main question is whether there is a main effect of condition, and, more importantly, whether condition interacts with the linear and/or quadratic effects of Bin. So, the results should focus on these interactions.

Response: Done.

Organizing the information in terms of learning performance and information-seeking behaviors should be reflected in the introduction, methods, and discussion, as well as in the results section. Be sure to use consistent terms throughout the paper for “information-seeking behaviors” – sometimes they are referred to as information access, information sampling strategies, information sampling, or attentional optimization.

Response: We have settled on “information access behaviors” as the preferred term and have made additional effort to distinguish this from our measure of attentional optimization.

Other suggestions follow:

a. Include a photo or drawing of the controllers in the three conditions.

Response: We have added photos of the controllers used in each condition: Fig 3

b. What are the outcome measures used in the Hamilton et al. meta-analysis? (line 152)

Response: We have made the quantitative outcome measures reported in this meta analysis more explicit.

c. The comparison group is missing in several places (e.g., measurable improvements over what?) (lines 153, 165, 178, 186, 226, 237, 243)

Response: Thank you for pointing this out. We have updated these lines to be clearer.

d. Add the DVs to Conditions in the Methods section (line 290); for example, “Conditions and Outcome Variables” but save how they were calculated for later in the methods section, in a section called “Calculation of Dependent Measures” (or something like that). “Detecting Fixations” would become part of this new section.

Response: Done.

e. The paragraph describing a trial (lines 342-353) is still confusing. It should be in the form: A trial began with_____, then the stimulus appeared, and the participant manipulated the object (VR and 3D conditions) or simply looked at the stimulus (2D condition). When they decided which category the stimulus belonged to, they _____. After responding, they received feedback until they were ready for the next trial, at which time they pressed ____. This produced a arrow with the words “next trial.” When the participant pressed ____, the next trial began.

Response: We have restructured our procedures description to be more concise using aspects of the template provided by the reviewer here.

Reviewer #2: 

Most of my suggestions have been responded to in the manuscript and commented on in the cover letter. The structure of the paper has been improved, and the description of the methodology has been improved. I do not understand why the authors insist on naming the eye-tracking ("ET") condition. J still believe that this tested variant should be marked as "2D".

Response: We have come (albeit slowly!) to agree with you and the other reviewer regarding the naming of this variable and have changed the name in the manuscript to 2D.

Finally, in your response to our initial and most recent revisions, Reviewer #2 indicated that our data is inaccessible online. However, our data availability statement clearly indicates where you can access our publicly released data in .csv format. This link works for us. If you are having any issues accessing this data, please let us know and we can troubleshoot appropriately.

---

## [Decision Letter · Decision Letter 2]

12 Sep 2022

Comparing Virtual Reality, Desktop-Based 3D, and 2D Versions of a Category Learning Experiment

PONE-D-22-02993R2

Dear Dr. Barrett,

We’re pleased to inform you that your manuscript has been judged scientifically suitable for publication and will be formally accepted for publication once it meets all outstanding technical requirements.

Kind regards,

Hong-jin Sun, Ph.D.

Academic Editor

PLOS ONE

Additional Editor Comments (optional):

Reviewers' comments:

Reviewer's Responses to Questions

**Comments to the Author**

1. If the authors have adequately addressed your comments raised in a previous round of review and you feel that this manuscript is now acceptable for publication, you may indicate that here to bypass the “Comments to the Author” section, enter your conflict of interest statement in the “Confidential to Editor” section, and submit your "Accept" recommendation.

Reviewer #1: All comments have been addressed

2. Is the manuscript technically sound, and do the data support the conclusions?

Reviewer #1: (No Response)

3. Has the statistical analysis been performed appropriately and rigorously? 

Reviewer #1: (No Response)

4. Have the authors made all data underlying the findings in their manuscript fully available?

Reviewer #1: (No Response)

5. Is the manuscript presented in an intelligible fashion and written in standard English?

Reviewer #1: (No Response)

6. Review Comments to the Author

Reviewer #1: Thank you for responding positively to my comments. The manuscript is much clearer, and I have no further suggestions.

7. PLOS authors have the option to publish the peer review history of their article (what does this mean?). If published, this will include your full peer review and any attached files.

Reviewer #1: **Yes: **Beverly Roskos

---

## [Editor Report · Acceptance letter]

16 Sep 2022

PONE-D-22-02993R2 

Comparing Virtual Reality, Desktop-Based 3D, and 2D Versions of a Category Learning Experiment 

Dear Dr. Barrett:

I'm pleased to inform you that your manuscript has been deemed suitable for publication in PLOS ONE. Congratulations! Your manuscript is now with our production department. 

Kind regards, 

on behalf of

Dr. Hong-jin Sun 

Academic Editor

PLOS ONE